# Upregulated Matrisomal Proteins and Extracellular Matrix Mechanosignaling Underlie Obesity-Associated Promotion of Pancreatic Ductal Adenocarcinoma

**DOI:** 10.3390/cancers16081593

**Published:** 2024-04-21

**Authors:** Richard T. Waldron, Aurelia Lugea, Hui-Hua Chang, Hsin-Yuan Su, Crystal Quiros, Michael S. Lewis, Mingtian Che, V. Krishnan Ramanujan, Enrique Rozengurt, Guido Eibl, Stephen J. Pandol

**Affiliations:** 1Karsh Division of Gastroenterology and Hepatology, Department of Medicine, Cedars-Sinai Medical Center, Los Angeles, CA 90048, USA; 2Department of Medicine, David Geffen School of Medicine, University of California, Los Angeles, CA 90095, USA; 3Department of Biomedical Sciences, Cedars-Sinai Medical Center, Los Angeles, CA 90048, USA; 4Department of Medicine and Department of Pathology & Laboratory Medicine, VA Greater Los Angeles Health System, Cedars-Sinai Medical Center, Los Angeles, CA 90073, USA; michael.lewis3@cshs.org; 5Biobank and Research Pathology Resource, Cedars-Sinai Medical Center, Los Angeles, CA 90048, USA; 6Department of Surgery, David Geffen School of Medicine, University of California, Los Angeles, CA 90095, USA

**Keywords:** pancreatic ductal adenocarcinoma, KC mice, diet-induced obesity, quantitative proteomic analysis, tandem mass tags, extracellular matrix, matrisomal proteins, tumor microenvironment, prognostic biomarkers, atypical flat lesions

## Abstract

**Simple Summary:**

Epidemiologic evidence and previous research have established obesity, diabetes and the metabolic syndrome as risk factors for pancreatic cancer in humans, while the precise mechanisms remain incompletely characterized. The concept that these metabolic diseases may be driven by diet implies that many cancers could be prevented or delayed through lifestyle modifications. By design, genetically engineered mouse models expressing oncogenic mutants of KRAS protein within the epithelial cells of the pancreas closely recapitulate the human disease. Research on diet-induced obesity in such mice implies the involvement of diverse factors including elevated insulin/insulin-like growth factors, inflammation, and genetic/epigenetic alterations and has revealed dramatic acceleration of cancer incidence that was more marked in males than females. The present research uses proteomic and phosphoproteomic analysis as a foundation to assess differentially expressed proteins and associated signal transduction pathways, yielding further novel mechanistic insights into the influence of tumor microenvironment on diet-induced obesity-associated pancreatic cancer.

**Abstract:**

Diet-induced obesity (DIO) promotes pancreatic ductal adenocarcinoma (PDAC) in mice expressing KRasG12D in the pancreas (KC mice), but the precise mechanisms remain unclear. Here, we performed multiplex quantitative proteomic and phosphoproteomic analysis by liquid chromatography–tandem mass spectrometry and further bioinformatic and spatial analysis of pancreas tissues from control-fed versus DIO KC mice after 3, 6, and 9 months. Normal pancreatic parenchyma and associated proteins were steadily eliminated and the novel proteins, phosphoproteins, and signaling pathways associated with PDAC tumorigenesis increased until 6 months, when most males exhibited cancer, but females did not. Differentially expressed proteins and phosphoproteins induced by DIO revealed the crucial functional role of matrisomal proteins, which implies the roles of upstream regulation by TGFβ, extracellular matrix-receptor signaling to downstream PI3K-Akt-mTOR-, MAPK-, and Yap/Taz activation, and crucial effects in the tumor microenvironment such as metabolic alterations and signaling crosstalk between immune cells, cancer-associated fibroblasts (CAFs), and tumor cells. Staining tissues from KC mice localized the expression of several prognostic PDAC biomarkers and elucidated tumorigenic features, such as robust macrophage infiltration, acinar–ductal metaplasia, mucinous PanIN, distinct nonmucinous atypical flat lesions (AFLs) surrounded by smooth muscle actin-positive CAFs, invasive tumors with epithelial–mesenchymal transition arising close to AFLs, and expanding deserted areas by 9 months. We next used Nanostring GeoMX to characterize the early spatial distribution of specific immune cell subtypes in distinct normal, stromal, and PanIN areas. Taken together, these data richly contextualize DIO promotion of Kras-driven PDAC tumorigenesis and provide many novel insights into the signaling pathways and processes involved.

## 1. Introduction

Pancreatic ductal adenocarcinoma (PDAC), a devastating disease with high mortality and limited treatment options, is increasing in incidence in parallel with risk factors such as diabetes and obesity. Information on the tumorigenic mechanisms that mediate such effects has been sought to improve both treatment and strategies to prevent PDAC development. It is recognized that activating KRAS mutations drives tumor initiation and progression in the vast majority of human PDAC [1,2]; this process also requires a second major stimulus, such as inflammation [3]. Epidemiologic data strongly implicated diet-induced obesity (DIO) as a risk factor in human PDAC [4]. Our previous studies demonstrated that DIO accelerates PDAC tumorigenesis in genetically engineered mouse models [5]. Mice expressing oncogenic mutant KRasG12D in the pancreas (KC mice) and fed a high-fat, high-calorie diet (HFCD) to approximate a Western diet rapidly became obese and exhibited dramatic acceleration in PDAC development and a much higher incidence at 6 and 9 mo of age compared to KC mice on a control diet (CD) [6]. Invasive tumors formed especially in males at 6 mo of age and more equally in older males and females. These mice recapitulate major aspects of human PDAC development and represent a valuable model to further investigate the mechanisms of DIO-induced PDAC. As highlighted in recent reviews [7,8], data from this model point to some central mechanisms of DIO-induced tumorigenesis. Inflamed, “unhealthy obese” adipose depots and accompanying increased localization of immune cells and associated inflammatory processes near the pancreas appear critically important [9,10]. Also, metabolic conversion of lipids in the high-fat diet into bioactive molecules may enhance KRAS signaling [11] and/or limit DNA repair in tumor cells. For example, mevalonate synthesis supports cholesterogenesis, a key pathway that is a novel hallmark of PDAC [12,13] and provides precursors for the protein prenylation of signal transducers.

Our previous studies associated DIO in KC mice with metabolic changes, visceral adipose deposition, inflammation, elevated plasma insulin/IGF, cytokines and chemokines, high somatic mutation rates, and higher emergence and progression rates of preneoplastic pancreatic intraepithelial neoplasia (i.e., PanIN) lesions [5,6]. Eventually, KC mice with DIO showed higher incidence of aggressive, invasive tumors compared to the CD-fed KC mice [6]. Western blotting pancreas tissues from these mice revealed disrupted autophagy, and RNA sequencing uncovered mutations in relevant pathways such as the PDGF receptor and PI3K/mTORC-mediated signal transduction [6]. Here, we investigate further the tissue changes in HFCD-fed KC mice using high-sensitivity liquid chromatography–tandem mass spectrometry (LC-MS/MS) with quantitative proteomic and phosphoproteomic approaches to gather additional evidence of the proteins, phosphoproteins, and pathways underlying the protumorigenic effects of DIO in PDAC.

We also used complementary quantitative histochemical staining and immunohistochemical localization in pancreas tissues from KC mice to substantiate findings on the role of DIO to accelerate PDAC tumorigenesis, provide additional information on the protumorigenic roles of microenvironmental factors, and establish specific markers to distinguish more aggressive from relatively benign cellular phenotypes. We also characterized spatial distributions of immune cell subtypes between normal, PanIN, and stromal regions at early stages of DIO-promoted PDAC tumorigenesis using Nanostring GeoMX. Our analysis builds a more complete picture, highlighting the roles of specific signal transduction pathways and gene expression networks that mediate DIO-induced PDAC tumorigenesis.

## 2. Materials and Methods

### 2.1. Mouse Model and Pancreas Homogenization

Genetically engineered mice expressing a mutant *KrasG12D* transgene in the pancreas (*Kras*^LSL.G12D/+^; *Ptf1*^Cre/+^; termed KC mice) were established as a model system to study the role of mutant KRas as an important initiator of PDAC [2]. Here, four-week-old mice were randomly allocated to a control diet (CD) or a high-fat, high-calorie diet (HFCD). Pancreas tissue was harvested from mice euthanized at three, six, and nine months of age and homogenized on ice using a Glas-Col Tissue homogenizer with conical Kontes Duall 23 glass tubes and an HPDE pestle with a steel handle. Approximately 25 mg of each tissue was homogenized in 500 µL of buffer using 20 up-strokes of approximately 20 s duration, with 30 s resting on ice between bursts. The buffer comprised 100 mM Triethylammonium bicarbonate, pH 8.5, supplemented with Roche (Bsel, Switzerland) PhosSTOP phosphatase inhibitor cocktail and 2 mM sodium orthovanadate. Homogenates were quick-frozen and stored at −80 °C. Protein concentrations were measured by a BCA microassay.

### 2.2. Proteomic Sample Preparation and TMT Labeling

Filter-assisted sample preparation [14] was carried out with the KC mice pancreas homogenate samples as follows: 100 µg of each sample was extracted by boiling for 5 min in 100 µL of 2% SDS, 50 mM TEAB buffer followed by probe sonication for 1 min at high speed, then spun at 10,000× *g* in a microfuge at room temperature (RT) and the supernatant was transferred to a 30,000 MWCO cellulose acetate spin filter module. The reducing agent, TCEP, was added to 10 mM, and the tubes were placed on a ThermoMixer and heated to 55 °C for 1 h with occasional mixing. Then, 8 M urea in 50 mM TEAB (UT) buffer (300 µL) was added, the samples were mixed, and buffer containing excess reducing agent and SDS was depleted from the sample by centrifuging the soluble components through the filter bed at 10,000 rpm on a microfuge at RT. When samples in the filters reached ~50 µL, they were washed again with 300 µL UT buffer. To alkylate proteins, 50 mM iodoacetic acid was added. The samples were mixed in the Thermomixer for 1 min, then placed in the dark for 30 min. One wash with UT buffer was followed by two washes with 50 mM TEAB buffer (200 µL each). Proteomic-grade trypsin (Thermo Fisher, Waltham, MA, USA) was dissolved in the buffer provided (20 µL), then combined with 50 mM TEAB, and 85 µL of this solution was transferred to each filter to give a final trypsin/protein ratio of 1:40. The filters were sealed with Parafilm, then shaken at 600 rpm and 37 °C for 18 h.

Following trypsinization, peptides were extracted from the filters by adding 40 µL 50 mM TEAB twice, shaking at 800 rpm for 20 min at 37 °C, and then spinning the filter tubes to collect peptide samples in clean LoBind tubes. A final extraction was performed using 50 µL of 5 mM NaCl in 50 mM TEAB. All peptides spun through the filter were combined and the volume reduced to ~170 µL by vacuum centrifugation. Equal portions of each sample were combined into a “master mix”, or reference standard, sufficient for four nine-plexes. Peptides derived from pancreas tissue from CD- and HFCD-fed mice were labeled with different TMT labels according to the manufacturer’s instructions. Four nine-plexes corresponding to pancreas tissue harvested at either 3, 6, or 9 mo were assembled, containing equal amounts of labeled peptides from CD- and HFCD-fed male and/or female mice and the reference standard. Thus, three- and nine-month mixtures contained peptides from both male and female CD- and HFCD-fed mice, whereas at six months there was one mixture containing CD- and HFCD-fed pancreatic peptides of male mice, and another of female mice.

Freshly assembled nine-plex TMT-labeled peptide/phosphopeptide mixtures were desalted using OASIS HLB columns by loading in 1 mL 0.1% formic acid (FA) and washing ten times with 1 mL, then eluting once each with 30%, 50%, and 70% acetonitrile (ACN). ACN and volatile buffer were removed from the combined eluates by vacuum centrifugation. One-third (~300 ug) was designated for proteomic analysis. The other two-thirds (~600 ug) were used to perform phosphopeptide isolation by sequential IMAC and TiO_2_ column chromatography (Thermo Fisher Scientific) using kit components in spin columns according to the manufacturer’s instructions. Next, to generate proteomic and phosphoproteomic samples, the peptide mixtures were fractionated using high pH reverse phase spin columns (Thermo Fisher Scientific). Labeled peptides were added to the columns in a loading solution (LS) containing 0.1% FA/5% ACN, and the flow through was kept as fraction one. Then, two washes were performed with LS, and peptides were eluted in seven sequential steps using increasing amounts of ACN. The eight fractions arising from each nine-plex mixture were concatenated into three samples for LC-MS/MS. The resulting (12) concatenated labeled peptide mixtures were desalted using disposable C18 spin columns and stored at −20 °C. Similarly, isolated phosphopeptide nine-plexes were also fractionated and catenated into (12) peptide samples and stored until analysis. 

### 2.3. Proteomic and Phosphoproteomic Analysis of TMT Nine-Plexes by LC-MS/MS and Systems Analysis

Tryptic peptide samples were analyzed using an Orbitrap Fusion Lumos (Thermo Fisher) ultra-high-resolution mass spectrometer equipped with Easy-nLC-1200 liquid chromatograph for LC-MS/MS at the Mass Spectrometry and Biomarker Discovery Core at Cedars-Sinai Medical Center. The data were processed by Mascot/Proteome Discoverer (v2.1) and then by MaxQuant (v.1.6.6.0). Protein IDs with at least 2 unique peptides and a false discovery rate of 1% at both peptide and protein levels were considered discovered and were further analyzed. Intensities were normalized across runs as described [15]. Briefly, TMT-labeled peptide intensity data obtained from Maxquant were summed across each run. The intensity values across each run were multiplied by a factor to normalize all the runs to an equivalent intensity. Normalized data were analyzed by Perseus software (Version 1.6.2.3) to quantitate the differential expression of identified proteins that increased or decreased by fold-changes in the HFCD/CD ratio and to perform Student’s *t* Test (*p* < 0.05 was used as a cutoff for statistical significance). Phosphopeptide samples were analyzed in the same manner as bulk peptides, and their levels were normalized to the relative amounts of total proteins.

### 2.4. Pathway Analysis and Bioinformatics Tools

Pathway searches were performed using the Database for Annotation, Visualization, and Integrated Discovery (DAVID) Bioinformatics Resource (https://david.ncifcrf.gov/) [16,17], (accessed on 15 May 2022) and web tools developed by the Gene Ontology results from DAVID were prepared for visualization using the REVIGO tool available at http://revigo.irb.hr/ (accessed on 18 October 2022). Further systems analysis of upstream regulators and downstream effects was carried out via the input of a list of elevated and reduced proteins, with associated fold-changes, into Ingenuity Pathway Analysis (IPA) software (QIAGEN Inc., Hilden, Germany, https://www.qiagenbioinformatics.com/products/ingenuity-pathway-analysis) (accessed on 17 October 2023). We also used Gene Set Enrichment Analysis (GSEA v.4.3.2, available at https://www.gsea-msigdb.org) (accessed on 18 February 2023) to evaluate gene enrichments in selected datasets. scRNAseq data from 9 PDAC datasets were extracted interactively from the Tumor Immune Single-cell Hub 2 (TISCH2) website (http://tisch.comp-genomics.org/) (accessed on 16 October 2023), as described in the text. Phosphosite (www.phosphosite.org) was also accessed to categorize phosphopeptides using the Motif Analysis tool and to predict upstream kinases using The Kinase Library webtool beta version as described in the text.

### 2.5. Histochemical and Immunocytochemical Staining of Fixed Pancreas Tissue, Spatial Analysis Using QuPath Software, and Spatial Characterization by Nanostring GeoMX

Murine pancreas tissues were paraffin-embedded, cryosectioned at 4 um thickness, and attached to slides. The slides were placed on a 60 °C heat block for 30 min, then plunged into a xylene series (5′-5′-10′) to deparaffinize and an alcohol series (100% twice, then 95–85–70%, 3 min each) to rehydrate, then stained with either Hematoxylin and Eosin (H&E), Masson’s Trichrome, Alcian Blue with a Nuclear red counterstain, or periodic acid–Schiff–Alcian Blue staining according to the manufacturer’s instructions. For immunohistochemical localization of most target proteins, slides were subjected to heat-induced epitope recovery in a steam cooker for 30 min with a 30 min cooling period using pH 6 citrate buffer supplemented with 0.05% Tween-20 prior to overnight incubation at 4 °C with primary antibodies. In contrast, for F4/80 and CD163 staining, epitopes were recovered by incubating rehydrated tissue with Proteinase K for five min in TE buffer at RT, as this method was more effective for these targets. Slides were blocked and permeabilized by 30 min incubation with 1% Aurion BSA-c acetylated albumin in PBS with 0.02% Triton X-100. Subsequent incubations used the blocking buffer without Triton X-100.

A Vectastain ABC kit (Vector Laboratories, Newark, NJ, USA) was used to amplify the biotin-conjugated secondary antibody and HRP-conjugated streptavidin binding steps. A signal was generated using an ImmPACT DAB Substrate kit. The tissue was incubated with DAB reagent for up to 5 min, washed extensively with PBS, counterstained with Gill hematoxylin for 10 min, and washed for 2 min in flowing tap water. A drop of VectaMount permanent mounting medium and a cover slip were placed over the stained tissue. Slides were imaged by the Cedars-Sinai Biobank and Research Pathology Resource at 20× magnification using an Aperio AT2 Slide Scanner system. Spatial analysis was performed using QuPath software [18] as we recently reported [19] and according to the instructions provided at the website, https://qupath.github.io (accessed on 21 March 2023).

### 2.6. Digital Spatial Profiling of Immune Cell Subtypes by Nanostring GeoMX

For Nanostring GeoMX Digital Spatial Profiling, we used the following commercial panels: the Background and Housekeeping protein panel, the GeoMx^®^ Immune Cell Profiling Panel, the GeoMx^®^ Immune Cell Typing Panel, and the GeoMx^®^ Immune Activation Status Panel. Briefly, selected slides from CD- and HFCD-fed male mice at 3 and 6 mo (*n* = 3) were deparaffinized as above. To select the regions of interest (ROIs), initial immunofluorescence staining was performed with anti-pan-cytokeratin for identifying epithelial cells, anti-CD45 for immune cells, and DAPI for nuclear blue stain. ROIs were selected upon visualizing the IF staining and guided by observing parallel H&E-stained slides of the same tissues. Eight ROIs per slide were selected as normal, PanIN, or stromal regions (in a 2:3:3 ratio). Then, the slides were incubated with reagents from the 4 selected GeoMx^®^ panels including primary antibodies conjugated to UV-cleavable oligonucleotides. Excess antibody was washed away, and the detection of antibody binding was achieved by illuminating the ROIs with UV, then collecting and depositing the oligonucleotides in a 96-well plate by a microfluidic device. A DSP Nanocounter digital analyzer system was used to document the oligonucleotide numbers for probe quantification, and data were analyzed by multivariate analysis using SPSS software version 24.0.0.0.

## 3. Results

### 3.1. Diet-Induced Obesity Promotes Mucinous PanIN, Fibrotic Microenvironment, and Altered Cellular Phenotype Culminating in Pancreatic Tumorigenesis in KC Mice

We previously reported that DIO promotes PDAC tumorigenesis in KC mice [5,6,19]. Here, the histologic (H&E) images in Figure 1 further illustrate the DIO-induced acceleration of PDAC tumor development in male and female mice over time, with dramatic features including the loss of acinar tissue and the advancement of a fibroinflammatory stromal TME. We previously counted pancreatic intraepithelial neoplasia (PanIN) to demonstrate DIO-induced advancement of these important precursor lesions [5,6]. Other investigators have previously used mucin staining by Alcian Blue (AB) to identify PanIN lesions [2,20,21]. Here, to briefly revisit DIO-induced neoplastic precursor progression, we quantified classic mucinous PanIN lesions on a cellular basis using QuPath spatial analysis software [18,19]. The data shown in Figure 2A illustrate that DIO promoted an increase in the number of PanIN cells at 3 mo and 6 mo. The data from males and females were combined to generate the aggregated data in Figure 2B. While only a few (~1%) of the cells were AB+ at 3 mo in CD-fed mice, DIO enhanced AB+ levels ~2-fold at this time point. By 6 mo, the number of AB+ cells in CD-fed KC mice had significantly increased to ~4%, and DIO had enhanced AB+ cells to nearly 8%, consistent with our previously published results demonstrating DIO-accelerated PanIN progression [5,6].

We previously stained collagenous ECM with Picro Sirius Red to examine the DIO-induced promotion of stromal expansion in PDAC [5,6]. Here, we used Masson’s Trichrome staining and QuPath to localize and quantify stromal collagen levels in representative pancreas tissues from female and male KC mice, which did not exhibit striking differences. Trichrome-stained pancreas tissues are shown in CD- and HFCD-fed KC mice at 6 mo (Figure 2C) and in CD-fed nontumorous tissue and HFCD-fed tumor tissue at 9 mo (Figure 2D). Quantification of Trichrome staining showed a large DIO-induced increase in collagen accumulation in thick interlobular bands and adjacent to ADM/PanIN at 6 mo, and a smaller increase at 9 mo (Figure 2E).

We also used H&E staining and QuPath to characterize and quantify major cell types arising during DIO-induced PDAC. We trained QuPath to recognize acinar, islet, immune, stromal cells/fibroblasts, acinar–ductal metaplasia (ADM), and PanIN. QuPath was also trained on cells populating cancer tissues lacking lobular structures identified by a pathologist. ADM contains dedifferentiated acinar cells, which may either regenerate the pancreas or convert into preneoplastic lesions [22,23]. The resulting images in Appendix A, summarized in Appendix A, show that DIO dramatically reduced acinar cells and increased tumor cells in the pancreas of KC mice. DIO also increased distinct low-cellularity stromal regions, immune cells, and PanIN structures, decreased ADM, and left islet cells unchanged. Cell counting at different times (Appendix A) and in females vs. males (Appendix A) confirmed major DIO-induced tumorigenic changes consistent with the scoring systems we used previously [5,6].

### 3.2. Proteomic Analysis Reveals Association of DIO-Induced PDAC with Elevated Matrisomal Proteins

The data in Figure 1, Figure 2 and Appendix A further confirm DIO-induced PDAC development in KC mice, and recapitulate the roles of PanIN, fibrosis, and altered cellular phenotypes in PDAC progression, but do not identify major changes in protein expression underlying these events. To examine specific DIO-induced increases and decreases in proteins accompanying PDAC development, we performed quantitative proteomic analysis and compared HFCD-fed to control-fed KC mouse pancreases. The proteins discovered here with at least two unique peptides from both female and male KC mice at 3, 6, and 9 mo are listed in Appendix A. We plotted the log expression levels of all proteins as the ratio of HFCD/CD to show DIO-induced, differentially expressed proteins, as illustrated in Figure 3A. DIO significantly (≥1.5-fold, *p* < 0.05) induced the overexpression of 188 proteins, and suppressed (≤0.67-fold, *p* < 0.05) 141 proteins. The volcano plot in Figure 3B illustrates selected highly altered and significant proteins. Prominent reduced proteins included secreted digestive enzymes; Reg2, a Reg family member that protects from diabetes [24]; and Serpini2/Pancpin, a serpin family member crucial to maintaining acinar cell secretory function and identity [25]. Several matrisomal proteins were increased, including the collagens Col4a2, Col7a1, Col12a1, and complement C1qa and C1qb; laminins Lama3 and Lamb3; the matrix metalloprotease Mmp7; the lipocalin family protein Lcn7/Tinagl1; and the laminin-binding protein Nidogen-/Nid1. A full 10 of the 17 most prominent overexpressed proteins were matrisomal. Thus, matrisomal proteins form a major component of the DIO-induced elevated proteins in KC mice.

We next performed GSEA analysis of DIO-upregulated proteins to understand the upregulated pathways, revealing enrichment in (KEGG) tight junctions, the regulation of actin cytoskeleton, integrin signaling, focal adhesion, ECM–receptor interaction, and the TGFβ signaling pathway, all with multiple matrisomal proteins in common and *p* ≤ 0.1. Interestingly, comparison of our proteomic dataset against the Naba Core Matrisome Gene Set shows clear enrichment involving 57 matrisomal proteins in DIO (Figure 3C). Investigating Gene Ontology (GO) pathways related to DIO-overexpressed proteins by DAVID analysis, we found highly promoted GO biological processes, including endodermal cell differentiation, cell adhesion, innate immune response, and ECM organization, summarized in the Revigo map in Appendix A. Again, all these processes involve matrisomal proteins, as do elevated related cellular compartments (proteinaceous ECM, collagen trimer, laminin-5 complex, and basement membrane), and molecular functions (fibronectin and integrin binding). Appendix A lists all the significant and most DIO-increased and -decreased proteins overall and corresponding DAVID analyses.

We also analyzed pathways DIO up- or downregulated in PDAC using IPA, which predicted a network illustrating the importance of matrisomal proteins in DIO-induced PDAC tumorigenic signaling. In the signaling network in Figure 3D, collagens and laminins participate in mechanotransductive ECM signaling (via Rho, actin cytoskeleton, and Fak, not shown) to elicit Yap/Taz/Tead-mediated transcriptional regulation of key secreted matrisomal proteins, Ccn2/Ctgf, Postn, Spp1, and Bmp, involved in intercellular communication. Key upstream regulatory factors derived from our proteomic data by IPA are shown in Figure 3E. Taken together, these features imply that DIO-promoted ECM proteins bind integrins and activate MAPK, PI3K, and other protumorigenic signal transduction pathways.

### 3.3. Analysis of DIO-Induced Matrisomal Protein Enrichment in Subgroups of KC Mice

Subgroup analysis of ~2500 proteins by time and sex in a two-way ANOVA resolved 1399 significant proteins elevated or decreased by DIO at 3, 6, and 9 mo of age in male and female mice, provided in Appendix A. DIO enhanced only a few proteins at 3 mo, when all mice exhibited PanIN, but only 10% (1/10) had cancer. Proteins elevated early with potential roles in tumor initiation include transferrin receptor (Tfrc), which alters mitochondrial metabolism and reactive oxygen species production [26], and thymosin-beta4 (Tmb4x), which may protect tumor cells from ferroptosis [27]. DIO induced the peak expression of many proteins (188 increased and 132 decreased) at 6 mo, when 3/10 (30%) of the HFCD-fed (i.e., 0/7 female but 3/7 = 42.9% male) mice had cancer. Fewer proteins were elevated (38) and decreased (116) at 9 mo, when 5/10, or 50%, of female and male mice had cancer, implying that 6–9 mo represented a peak time of DIO-induced tumorigenic changes in protein expression. As key proteins DIO selectively elevated or decreased might be important for sex differences in PDAC incidence (assessed by a pathologist [6]) at 6 mo, we performed unsupervised clustering of the most differentially expressed proteins in 6-month-old male and female KC mice, shown in Figure 3F. In this analysis, data from females (*n* = 6) clustered separately from the males (*n* = 6). We also performed separate subgroup analysis, shown in Appendix A, for the effect of DIO on cancer, in which 108 (highly enriched in matrisomal) proteins were significantly elevated and 122 proteins were downregulated, versus noncancer tissues.

Matrisomal proteins were mostly low at 3 mo, became enriched at 6 mo, and were sustained at high levels and associated with advanced tumors at 9 mo. Here, matrisomal protein enrichment measured in association with DIO-induced differentially overexpressed proteins in each subgroup increases in parallel to DIO-induced PDAC development (Appendix A). Thus, it is relatively low at 3 mo and in females, increases in males and at 6 mo, is even higher in cancer- vs. noncancer-bearing mice, and is highest at 9 mo, when females also contribute to tumor emergence, further highlighting the importance of ECM–receptor signaling in DIO-induced PDAC development in KC mice.

### 3.4. Phosphoproteins Associated with DIO-Induced PDAC Development in KC Mice

We also generated complementary phosphopeptide proteomic data. All phosphopeptides and their relative quantitation in response to DIO overall and in subgroups by ANOVA are shown in Appendix A. The most elevated phosphosite we found overall was ribosomal S6 Ser244, a prominent downstream target of p70S6K [28]. We also noted increases in phosphosites impacting intermediate filaments and actin cytoskeleton (Krt8-Ser43, Krt20-Ser11, Eppk1-Ser2705, Synpo-Ser833, Lcp1-Ser5), matrisomal proteins (Spp1-Ser234, Fbn1-Ser2704, Tagln2-Ser163), and regulating nuclear functions and mitosis (Npm1-Ser125, Srrm1-Thr913, Lmna-Ser404, Hp1a-Ser92). Acaca-Ser79 was most decreased overall, implying low AMPK activity [29], and was consistent with low fatty-acid synthesis in the DIO pancreas. Two sites in Yap1 targeted by Lats kinases in the Hippo tumor suppressive pathway, Ser94 and Ser112 (equivalent to Ser109 and Ser127 in human YAP1), were reduced, implying Yap1 activation in both males and females. These highlighted data and those in Appendix A broadly substantiate our bioinformatic analysis.

We next analyzed protein kinase substrate motifs of 25 top DIO-elevated phosphopeptides overall and in subgroups to produce the motifs shown in Figure 4A. We separated Proline-directed from non-Proline-directed motifs, as mainly distinct protein kinases target each kind of motif. Thus, groups that exhibited Proline at the S + 1 position, implying prominent targets of, e.g., MAPK and cyclin-dependent kinases belonging to CMGC and, e.g., mTOR to the other protein kinase family are shown in Figure 4A. Shown in Figure 4B, Arginine at the -3 position was prevalent in all the other motifs, corresponding to basophilic kinases of, e.g., the CAMK (CAMK, MARK, PKD3) and AGC (PKA, PKC, AKT) families. Some showed (-) charged residues at S + 2 and/or S + 3 corresponding to a CK motif, and the SXE motif favored by Golgi Casein kinase (CK) [30], also known as Fam20C, arose at 9 mo (Figure 4A). This motif analysis implicated the involvement of several protein kinases, such as ERK, JNK and P38 MAPK, AKT, CAMK, and CK family kinases, in DIO-induced PDAC. We also used the Kinase Library webtool associated with an Atlas of protein kinase substrate specificities [31] to illustrate the protein kinases DIO elevated and suppressed in our phosphopeptide data with a volcano plot, shown in Appendix A.

DAVID analysis of the parent proteins harboring DIO-increased phosphopeptides, shown in Appendix A, revealed biological processes, like cell adhesion, actin cytoskeletal signaling, and the regulation of cell migration, that further implicate matrisomal proteins in DIO-induced PDAC. Our phosphorylation data should enable further mining for novel hypotheses, as we uncovered many differentially expressed phosphopeptides with unknown biological significance.

### 3.5. Spatial Analysis to Determine Relationships between ECM Deposition, PDAC Initiation and Progression, and Distinct Responsible Cell Types

We next stained specific DIO-induced marker proteins, Muc4 and Maspin/Serpinb5, to determine their cell-type expression. We found these proteins were expressed predominantly in cells lining PanIN, as shown in Figure 5A. Thus, Muc4 and Maspin/Serpinb5 staining was coincident with that of AB+ cells lining the PanIN of control-fed mice at 9 mo. Many of these cells also stained for Agr2, a PDI family protein involved in mucin expression [32]; p-Erk-Tyr202/Thr204, a signaling kinase activated by protumorigenic signaling pathways [33]; and Yap1/Taz, transcriptional regulators important for PDAC initiation and progression [34] and for sustaining the activated state and fibroinflammatory responses of PaSC [35] in the tumor microenvironment. These markers mostly persisted in glandular structures within cancer tissue at 9 mo. Importantly, DIO selectively enhanced p-Erk and Yap/Taz staining (which also localizes to stromal cells), but not Agr2, which became more mosaic in its staining pattern, or Maspin and Muc4, which decreased (Figure 5A).

To explore marker expression of major ECM-depositing cell types in tissues with extensive Trichrome staining, we stained smooth muscle actin (α-Sma), i.e., Acta2, a marker of activated PaSC, myofibroblasts, and myCAFs [36]. Unexpectedly, we found that α-Sma+ cells were sparse within wide interlobular collagen bands and, instead, preferentially clustered in discrete, smaller Trichrome+ sub-areas. In case α-Sma+ myCAFs were not the only cells depositing stromal ECM, we also localized macrophages, which were recently implicated in strong ECM deposition after pancreas damage [37]. We found that the F4/80+ macrophage staining pattern was more like the thick interlobular collagen bands (see Figure 5B). In this case, distinct zones contain advanced tumors with extensive α-Sma+ cells, or PanIN areas with strong F4/80+ cells and Trichrome staining. Such data suggested that macrophages contribute strongly to ECM deposition, as proposed by others [38]. However, these measurements may not capture dynamic changes, such as α-Sma+ CAFs depositing large amounts of collagen and subsequently migrating away from deposit areas or dying.

### 3.6. Emergence of Atypical Flat Lesions and Substantiation of Cell Types and Marker Proteins by Immunohistochemical Staining during DIO-Induced PDAC Acceleration

Whereas α-Sma+ myCAFs populated only parts of the Trichrome-stained areas, it was important to further assess the subregions with highly clustered myCAFs. Our IHC indicated reactive stroma containing α-Sma+ cells sometimes associated closely with mucinous PanIN. However, we also noted thick concentrations of α-Sma+ cells around distinct nonmucinous epithelioid cell-based lesions, lacking staining for AB or Maspin, markers we associated with PanIN. Figure 5C shows one such lesion close to a PanIN lesion with contrasting staining patterns at 9 mo in a CD-fed mouse. We propose that nonmucinous, flat epithelioid lesions with irregular structures surrounded by α-Sma+ cells represent atypical flat lesions (AFLs), previously described by Esposito and colleagues [39,40]. The nearby PanIN lesion, by contrast, expresses AB+ mucin, stains Maspin+, and its milieu is richer in Trichrome+ collagen but contains fewer α-Sma+ myCAFs. Notably, cells of both PanIN and AFLs stained robustly for Yap/Taz, mostly in the nucleus (Figure 5C).

We chose additional DIO-induced matrisomal (Spp1, Lama3) and nonmatrisomal (Yap/Taz, Hmga2) protein markers associated with tumorigenesis and poor prognosis [41] to validate and localize by staining. Previous studies of PDAC development associated matrisomal protein expression with mechanoregulatory signal transduction converging on Yap1/Taz [42]. Assigning cellular origins to matrisomal proteins through species-specific proteomic analysis of human-to-mouse xenografts, Tian et al. [43] elucidated that both stromal and tumor cell-produced matrisomal proteins are associated with poor prognosis in PDAC. We also stained other DIO-elevated proteins with possible tumorigenic roles, Spp1, Lama3, Hmga2, Yap/Taz, and embedded collagen by Trichrome stain, expressed in emerging cancer tissue in HFCD-fed KC mice at 6 mo, as shown in Appendix A. We found α-Sma+ myCAFs mostly concentrated in a pattern consistent with AFLs, contrasting with the presumptive mesenchymal marker vimentin which stained with a broader, primarily stromal pattern. TISCH2 scRNAseq data from PDAC (http://tisch.comp-genomics.org/) (accessed 16 October 2023) imply that Vim is expressed across diverse cell types, including monocytes and other immune cells. Whereas integrin-signaling Lama3 [44] intensely stains clustered epithelioid cells as in AFLs, Spp1, a protein implicated in immune cell infiltration, stains across AFLs and additionally, stromal cells. Yap/Taz also stains both epithelial and stromal cells in this context (see Appendix A). The Human Atlas website confirms Yap1, Spp1, and Lama3 as unfavorable prognostic markers in PDAC. These results further substantiate the proposition that matrisomal proteins participate in tumorigenesis and represent prognostic markers and potential therapeutic targets.

Our data associated a DIO-elevated marker, Hmga2, with AFLs. The representative stained images of KC mice in Appendix A illustrate the DIO-induced increasing emergence of Hmga2+ cells at 3, 6, and 9 mo. The cumulative data in Appendix A show that DIO induced significantly higher Hmga2 levels in tissues including cancer at 6 and 9 mo than the control diet, where it only appeared in precursor lesions. Our histologic data also associated AFLs with the emergence of an aggressive, mesenchymal tumor type. The serial images shown in Appendix A include such a region adjacent to a PanIN area. Initial cuts stained with H&E show a Maspin+ PanIN containing an upper region with lobular separations, and a lower region lacking lobular separations but featuring flat, Maspin-open lesions resembling ductal luminal spaces. In deeper cuts, staining shows the PanIN region staining for AB, Agr2, nuclear p-Erk1/2, and residual cytoplasmic/membranous epithelial marker Ecad. In contrast, the lower region has extensive α-Sma, due to intermingled myCAFs, Yap/Taz, and Ki67, indicative of highly proliferative tumor cells, yet virtually no mucin, Ecad, Agr2, or p-Erk1/2 staining. These characteristics resemble Kras-independent escaper populations of highly malignant mesenchymal PDAC, which were reported to arise even in murine PDAC models [45]. Taken together, the data in Appendix A further document key DIO-induced target proteins important for PDAC diagnosis and prognosis and localize them to neoplastic cells and stromal cells within distinct emerging tumor phenotypes.

### 3.7. Immune Cell Profiling in PanIN and Stromal Areas during DIO-Induced PDAC by Nanostring GeoMX

To further characterize DIO-induced development of the PDAC tumor environment by its immune cells, we next explored the effects of DIO-induced proteome changes using Nanostring GeoMX to assess marker expression in normal, stromal, and PanIN regions in KC mice fed a CD and HFCD at 3 mo and 6 mo (see Figure 6A). As expected, at 3 mo, DIO induced the stromal markers α-Sma, fibronectin, and CD31 (an endothelial marker) in stromal regions, and PanCK faithfully marked PanIN regions, regardless of diet (Figure 6B,C). DIO elevated total immune cells, measured by CD45 expression, myeloids/macrophages (F4/80), and antigen-presenting cell markers (CD11b; CD11c) early in stroma. T cells (CD3e) and subsets (e.g., CD4+), initially present near PanIN in CD-fed mice, were depleted there by DIO, and elevated at 3 mo and 6 mo (Figure 6D) in normal and stromal areas. Some additional activated T cell markers were elevated at 3 mo (CD8), at 3 mo and 6 mo (CD127/IL7RA), or in stromal/PanIN areas with CD at 6 mo and were otherwise reduced (PD1; CD40L; CD44). DIO strongly diminished CD86, a marker of M1 macrophages in normal/PanIN regions at both 3 mo and 6 mo. Similarly, CD14, a monocyte/myeloid marker, was strongest in PanIN areas and increased in response to the HFCD at 3 mo and diminished at 6 mo. In contrast, DIO induced CD163, which marks alternatively activated M2 macrophages in PanIN regions at 6 mo, as shown in Figure 6. DIO also elevated the hematopoietic stem cell marker CD34 in stroma. GZMB, a marker of cytotoxic T cells, was elevated in PanIN regions by the HFCD at 6 mo. These data are consistent with DIO inducing M2 macrophages that play a major role in early PDAC development in KC mice.

Our GeoMX data, confirming expected Housekeeping proteins high in normal areas (S6), PanIN (histone H3), increased stromal markers (α-Sma, fibronectin) at 3 mo, and increased PanCK in PanIN regions at 6 mo, reflected DIO-induced acceleration of PDAC development. Data from immune markers showed significant immune infiltration of myeloid cells (monocytes, macrophages) and T cells into developing PDAC, as shown previously [46,47]. Here, we observed (mainly at 3 mo) that DIO was associated with decreased T cell marker CD3 and increased monocytes/M2 macrophages in PanIN areas, and continuously elevated T cells and macrophages in stromal areas. This pattern is consistent with an immunosuppressive environment adjacent to PanIN, recognized as a major mechanism for PDAC to progress while eluding antitumor immunity [46,47]. Thus, it was paradoxical to find GZMB elevated in PanIN regions with the HFCD at 6 mo, as this would suggest robust antitumor immunity, unless the enzyme was ineffectual. Another interesting finding was the elevated CD14 levels around PanIN, especially at 3 mo by the HFCD; this did not closely match other myeloid markers, or the CD45 expression pattern. Macrophages play roles in PDAC development [47]; CD14 signaling induced by lipopolysaccharide was reported to “retune” immunosuppressive M2 macrophages, converting them to a tumor-suppressive phenotype [48]. Figure 6E summarizes the proteins significantly elevated in PanIN and stromal areas.

## 4. Discussion

Here, using mostly alternative methods, we briefly revisited the DIO-induced acceleration of PDAC in KC mice to set the stage for further analysis. Thus, Figure 1 shows the histologic picture, Figure 2 the staining of markers of PanIN (AB) and fibrotic stroma (Trichrome), and Appendix A the cellular phenotypes that emerge during malignant transformation [5,6]. We previously reported that males developed DIO-induced PDAC faster and hypothesized that fat distribution plays a role in the higher cancer incidence in males [6]. However, this selectively accelerated DIO-induced PDAC incidence was not predicted by the markers, which largely tracked in parallel with weight gain in both sexes [6]. In Figure 2B, counting AB+ cells to identify early PanIN likewise showed a major influence of DIO, but we did not detect a large difference between sexes. Moreover, the difference in this early marker was mainly transient, because AB+ PanIN continued to emerge over time in CD-fed KC mice, whereas their mucin content peaked and then declined due to the contribution of large DIO-induced cancer regions. Similarly, the data in Figure 2E reflected that illustrated in Figure 2C, D, i.e., a major increase in collagen staining at 6 mo in response to DIO, but not further increases later, likely reflecting ECM remodeling in PDAC.

We next performed proteomic and phosphoproteomic analysis of pancreatic tissues using a systems biology approach to further investigate mechanisms of DIO-induced PDAC development in KC mice. A central finding described here for the first time is that DIO highly enriched matrisomal proteins overall. Whereas Tian et al. [43] enriched the ECM to achieve a 90% relative abundance of matrisomal proteins from human and mouse tissues, we show that DIO overexpresses matrisomal proteins in KC mice even without tissue fractionation. The concept of matrisomal proteins recently emerged to emphasize the ECM and related secreted proteins, classified as follows: collagens (Types I, III, IV, V, VI, and XV), proteoglycans, glycoproteins, ECM-associated annexins and galectins, and secreted and ECM regulatory proteins [11,43,49,50]. Matrisomal proteins operate in mechanotransduction networks involved in tumorigenesis [11,43,50,51,52,53,54,55]. The findings of key DIO-induced collagens, laminins and fibronectin and Tenascin-C [54] imply their actions as integrin receptors in ECM–receptor signaling. These prominent signaling pathways stimulate downstream FAK phosphorylation [56] leading to PI3K, MAPK, and Yap/Taz signaling to promote DIO-induced PDAC tumor cell growth and migration (as shown in Appendix A) [54]. It will be crucial to address the functional roles of these proteins in further research. Recently, integrin–ECM signaling was targeted by a rationally designed protein, called ProAgio, that was shown to selectively inhibit αvβ3- and αvβ5-mediated signaling and induce apoptosis by an allosteric mechanism [57]. ProAgio reduced stroma and prolonged survival in KPC mice [58]. Indeed, our initial experiments indicate that the administration of ProAgio during DIO-induced PDAC in KC mice reduces the loss of acinar cells and the emergence of neoplastic cells and slows stromal advancement in this model (Lihong Huo, Aurelia Lugea, Richard Waldron, Zhi-Ren Liu and Stephen Pandol, unpublished), supporting the notion that ECM–receptor signaling drives these processes.

Whereas we noted previously that PDAC emerged earlier in males than in females [6], here matrisomal protein expression was less enriched in females than in males. Our data are most consistent with DIO-induced tumorigenesis in female KC mice being delayed until around 9 mo, when they normally approach perimenopause and begin to transition to reproductive senescence [59]. At this time in female KC mice with DIO, we observed increased Saa1, an acute phase protein. We also found other notable differences in protein expression. For example, lipocalin 2 (Lcn2) and Lim domain only 7 (Lmo7) were higher in males at 6 mo (Figure 3). These proteins are implicated in PDAC initiation and progression [60,61], and while we have not investigated the mechanism, their differential expression in males and females was noted previously [62,63]. Further, we found progesterone receptor (Pgrmc1) Ser181 phosphorylation, which plays a role in estrogen receptor activation [64], selectively elevated in females (Appendix A). Proinflammatory signaling, associated with human menopause [65], suggests a possible hormonal basis for differences in DIO-induced PDAC between male and female KC mice ([6] and this study). Further studies are needed to clarify the functional underpinnings of sex differences in PDAC.

The tumorigenic driver mutant Kras alters cellular metabolism [1]. DIO enhanced the expression of genes supporting glycolysis, including hexokinase, glucose transporters, and lactate transporter isoforms (see Appendix A). Glycolytic gene expression was associated with HIF1α activated by hypoxia in the PDAC microenvironment [66]. The glucosamine biosynthetic pathway was also recently associated with elevated glycolysis in PDAC [66]. In Gfpt1, the key enzyme for UDP-GlcNAc synthesis and Ser261 or Ser243 phosphorylation in distinct human variants reduces the enzyme catalytic activity. We found the equivalent residue in mice, Ser259, less phosphorylated, implying activation in cancer versus noncancer tissues. Srebp transcription factors normally express lipogenic target genes including Fasn to convert dietary carbohydrates into fat [67]. Instead, here DIO reduced Fasn expression to very low levels. Despite this, we found that cholesterol biosynthesis genes, Hmgcs1 and Soat1, were elevated. Together, these findings suggest that DIO promotes preferential activation of Srebf2, but not Srebf1-mediated lipid synthetic gene expression [68]. This supports the beneficial effect of statin-based treatments, selectively targeting this pathway [69] as a precision medicine approach to PDAC. DIO-decreased molecules included propionyl CoA carboxylase b (Pccb), which breaks down branched-chain amino acids and fatty acids, and methylmalonyl-CoA mutase (Mut), which supplies succinyl-CoA to the TCA cycle. We also found enhanced pyruvate dehydrogenase inactivating phosphorylation, which is typically enhanced by tumorigenic stimuli and accumulated acetyl-CoA. Single cell RNAseq data extracted from TISCH2 (not shown) indicated that many of the DIO-elevated matrisomal proteins implicated in ECM–receptor signaling (Col4a1, Col4a2, Tnc, Lamb3, Itga2, Itgav, and Itga5) were expressed in fibroblasts, endothelial cells, immune cells, and malignant cells in human PDAC. Further studies should clarify these changes and identify the cell types responsible for the impact of DIO. Whereas our data thus far strongly imply that DIO accelerates PDAC development, evidence of unique tumorigenic mechanisms that otherwise would not occur remains elusive. Comparing pancreatic proteins elevated in CD-fed KC mice at 9 mo (not shown) with key DIO-elevated proteins, we found key differences. For example, the lipogenic molecules Fasn and Acly were selectively increased in the CD-fed mice, but not DIO overall (Appendix A). Interestingly, among the proteins increased in both were a few matrisomal proteins, such as Col6a1 and Col6a2, which are reported to be increased by interactions with a soft ECM [70]. In contrast, our data also show that DIO selectively elevates proteins such as Thbs2 (Appendix A) that are elevated by stiff matrices and associate with mechanosensitive signaling factors including Yap [70]. Thus, we propose that the promotion of tissue stiffness is a key factor in DIO-induced PDAC induction.

We also found phosphosites with roles in protumorigenic mechanoregulatory signaling related to the cytoskeleton, cell adhesion, ECM signaling, mitotic/nuclear/epigenetic regulation, and the emergence of key tumor cell phenotypes including unrestrained proliferation and invasion. Mining these data will reveal insights and therapeutic possibilities related to targeting both ECM/stroma and key tumorigenic signal transduction pathways in tumor cells.

PanIN development coincided with the staining of classic tumor marker proteins, SERPINB5/Maspin [71] and Muc4 [72]. PanIN-localized markers increased at 3 to 6 mo, but diminished in advanced aggressive, invasive cancers at 9 mo, consistent with the role of mucinous lesions as precursors during early stages of cancer development. Collagen, α-Sma, and F4/80 staining in cancer tissues of KC mice with DIO showed the roles of myCAFs and macrophages in ECM deposition. We found areas of significant intra- and interlobular collagen deposition in DIO-treated KC mice pancreas highly infiltrated with macrophages, but only clusters of α-Sma+ PaSC/CAFs. Thus, localized α-Sma+ PaSC/CAFs accounted for only part of the observed collagen deposition. Recently, Xue et al. found that pancreatic macrophages rapidly upregulated mRNA coding multiple collagen isoforms following pancreatic injury [37]. Our staining results support both PaSC/myCAFs and macrophages as important cell types contributing to pancreatic fibrosis after injury and in the ADM-PanIN-PDAC sequence, aligning with earlier reports [38].

Our IHC staining also found distinct precursor lesions, AFLs, that intensely stained for key DIO-induced matrisomal proteins Spp1 and Lama3 undertaking important roles in PDAC tumorigenesis. These proteins may prove useful as markers of early detection, distinguish PDAC from CP or other pancreatic diseases [73], or validate the emergence of malignant cells in PDAC. Whereas Spp1 is a crucial factor in neoplastic/stromal communication supporting immune cell invasion [74,75], the role of its phosphorylation status is less clear. DIO induced increased phosphorylation of several of the dynamically regulated sites. Also, in AFLs we detected that DIO promoted the expression of stromal regulated Hmga2 [76], a marker of metastatic PDAC. Hmga2-mediated epigenetic reprogramming may contribute to basal PDAC subtypes. Indeed, neoplastic–stromal crosstalk via matrisomal protein expression appears crucial for protumorigenic epigenetic regulation [49]. Whereas we found Ser100 and Ser104 phosphorylation in the Hmga2 C-terminal very low at 3 and 6 mo, Ser104 was elevated in cancer versus noncancer tissues. Hmga2 was recently reported to be dispensable for tumorigenesis in KPC mice [18]. However, as this finding was applied to a PDAC model driven ab initio by mutant p53, its precise relationship with malignant progression remains undefined. We also observed AFLs near rapidly growing, mesenchymal tumor areas staining negatively for p-Erk, suggesting they could be Ras-independent, but this is tentative as it was not statistically powered by our data.

Finally, multiplex Digital Spatial Profiling of proteins and RNA in fixed tissue has become an essential fixture in PDAC research [77,78]. Our GeoMX data used antibody panels to identify immune subtypes and activation statuses at 3 mo and 6 mo during DIO-induced PDAC development. Salient findings included DIO-elevated CD163+ M2 macrophages and T cell activation markers CD40L and CTLA4 in emerging PanIN regions. Despite this, general immune cell markers CD45 and F4/80 were relatively elevated in the stroma. These findings support an immunosuppressive role of M2 macrophages promoting PDAC development in obese mice, as hypothesized previously [79]. Our tissue staining and spatial quantitation experiments validated and clarified the meaning of key proteomic findings, supporting the importance of matrisomal proteins and distinct neoplastic precursors, PanIN, and AFLs in the pancreases of obese KC mice and providing valuable new insights into the mechanisms of DIO-induced PDAC tumorigenesis.

## Figures and Tables

**Figure 1 cancers-16-01593-f001:**
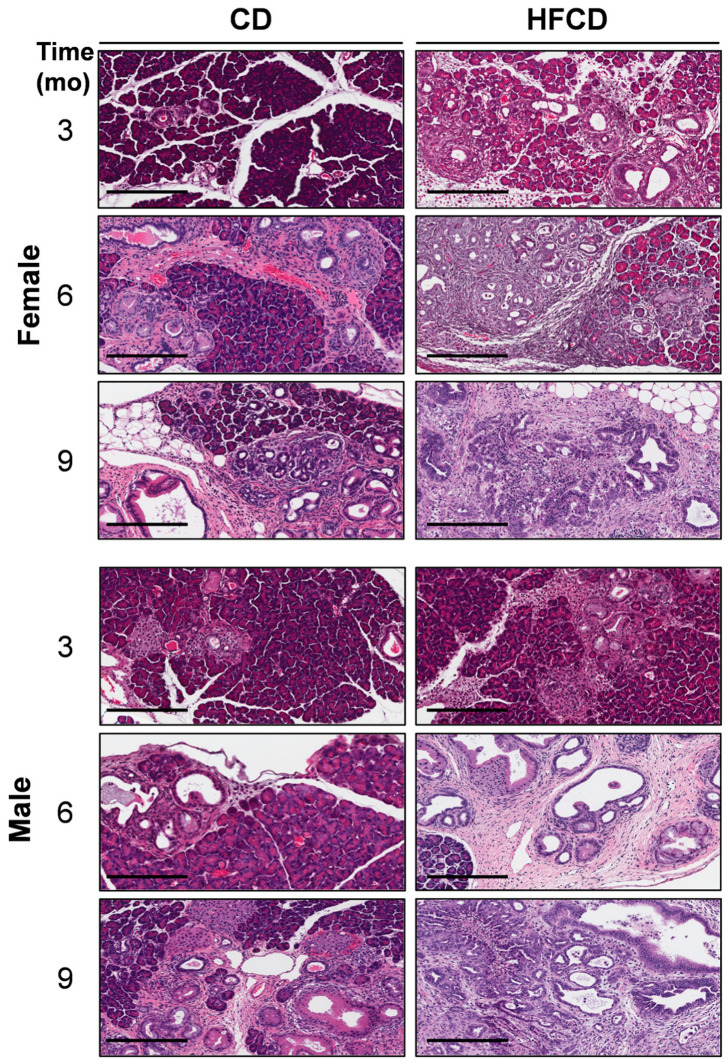
DIO-induced PDAC development. Representative pancreas histology (H&E) of PDAC development in female (**upper panels**) and male (**lower panels**) KC mice on a control diet (CD) versus a high-fat, high-calorie diet (HFCD) at 3, 6, and 9 months (mo) of age reveals extensive loss of acinar tissue concomitant with increased areas of fibrotic stroma, ADM, PanIN, and PDAC in HFCD-fed mice. Tumor tissues identified by a pathologist were selected to represent the effect of DIO in females at 9 mo and in males at 6 and 9 mo. Scale bar: 200 μm.

**Figure 2 cancers-16-01593-f002:**
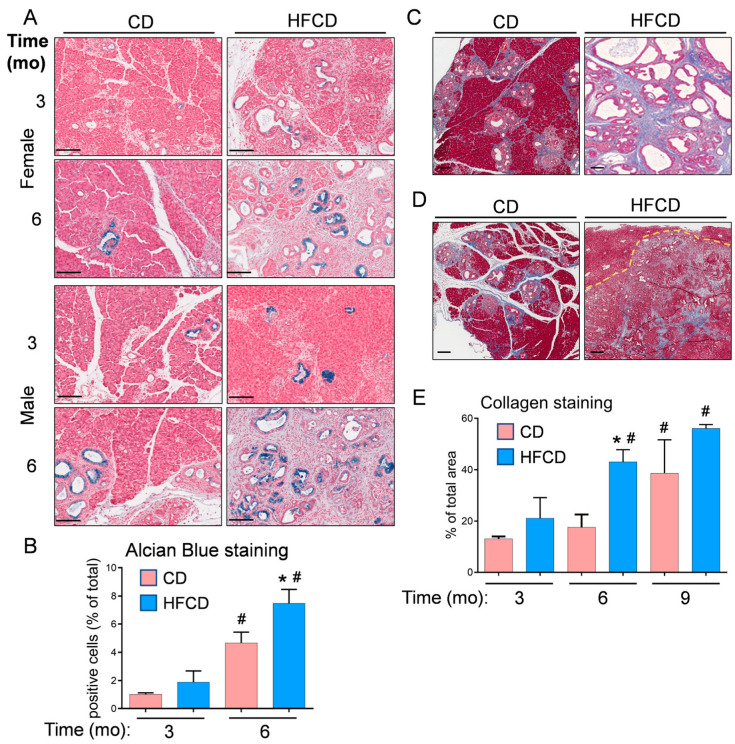
(**A**), representative Alcian Blue (AB) staining detects classical PanIN in the pancreas of KC mice fed a CD and HFCD. (**B**), quantitation of QuPath-classified AB+ cells in male and female pancreas tissues at different time points. (**C**), representative images of Trichrome staining at 6 mo show increases in collagen-stained areas in CD-fed and HFCD-fed KC mice. (**D**), stromal areas are also filled with ECM proteins in mice on a CD at 9 mo, and DIO induces smaller increases in mice exhibiting invasive pancreas/liver tumors. A dotted line demarcates a frontier between a tumor and an invaded liver area. (**E**), quantitation of a tissue area containing collagen stained blue with Masson’s Trichrome in KC mice on a CD versus an HFCD. The graphs in (**B**,**E**) show mean ± SEM. *, *p* < 0.05 vs. CD; #, *p* < 0.05 vs. 3 mo; ANOVA + post hoc Fisher’s LSD test. Scale bars, (**A**), 100 nm; (**C**,**D**), 200 nm.

**Figure 3 cancers-16-01593-f003:**
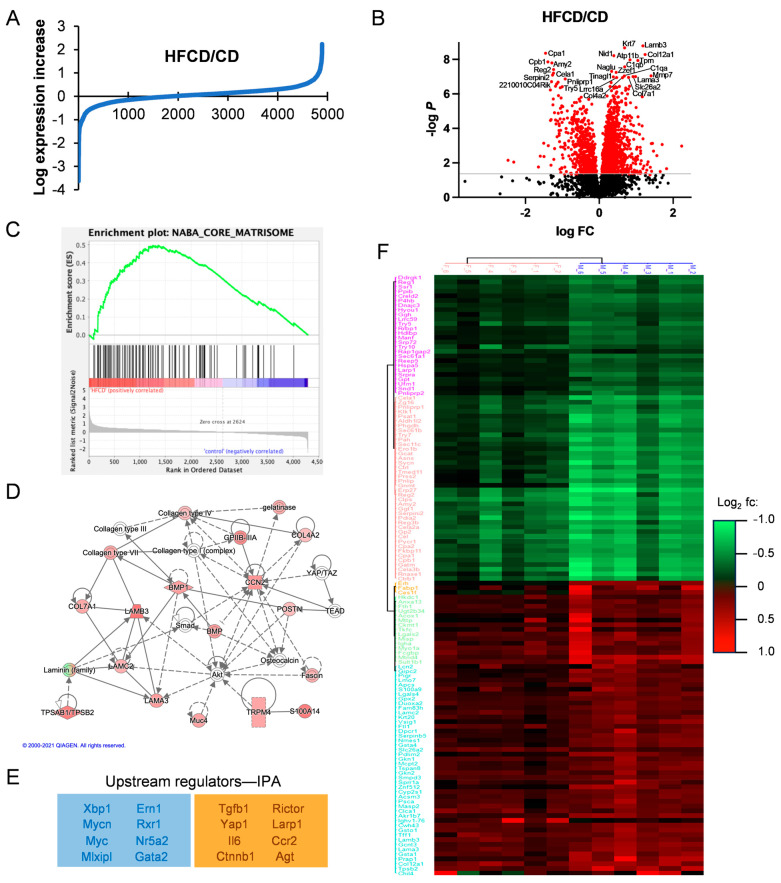
Matrisomal proteins support DIO-induced PDAC development. (**A**), log_2_-transformed HFCD/CD expression ratios. (**B**), volcano plot of HFCD/CD values with prominent down- (left) and upregulated (right) proteins labeled. (**C**), Gene Set Enrichment Analysis of HFCD- versus CD-fed mouse proteins within Naba Core Matrisome (see ref. [11]) gene set. (**D**), Ingenuity Pathway Analysis (IPA)-derived signaling network linking proteins upregulated by DIO with transcriptional regulators Yap1/Taz and discretely expressed matrisomal intercellular mediators such as Ccn2/Ctgf, Postn, and Bmp. (**E**), negative (blue box) and positive (orange box) upstream regulators of DIO-induced protein changes predicted by IPA. (**F**), unsupervised hierarchical clustering of proteins differentially elevated (red) or suppressed (green) in female (left) and male (right) KC mice by DIO at 6 mo. Protein names are shown on left. Legend indicates DIO-induced log2 fold change.

**Figure 4 cancers-16-01593-f004:**
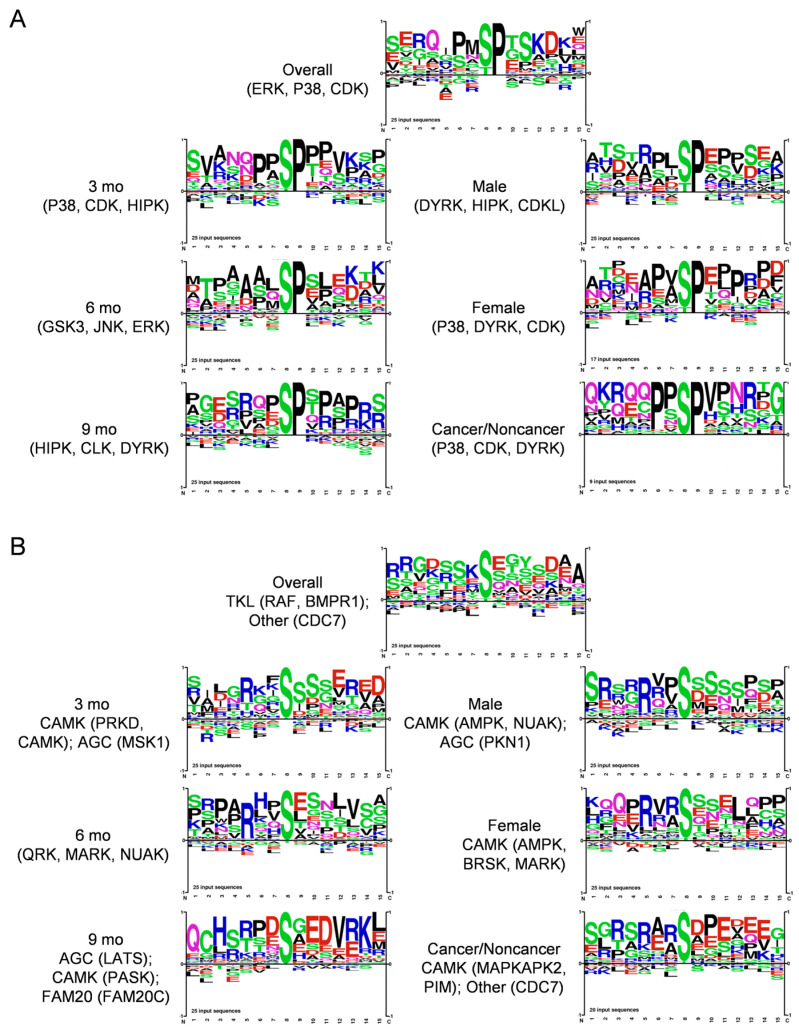
Motif analysis of 25 top (**A**), Proline-directed and (**B**), non-Proline-directed phosphopeptides that increased during DIO-induced PDAC induction in KC mice, using PhosphoSitePlus Tools. Each sequence logo depicts the preferred amino acid residues most found around the targeted Ser residues, with the preferred (larger) residues at each position stacked on top of lower-scoring ones. Kinases targeting the main sequences were also calculated and are shown.

**Figure 5 cancers-16-01593-f005:**
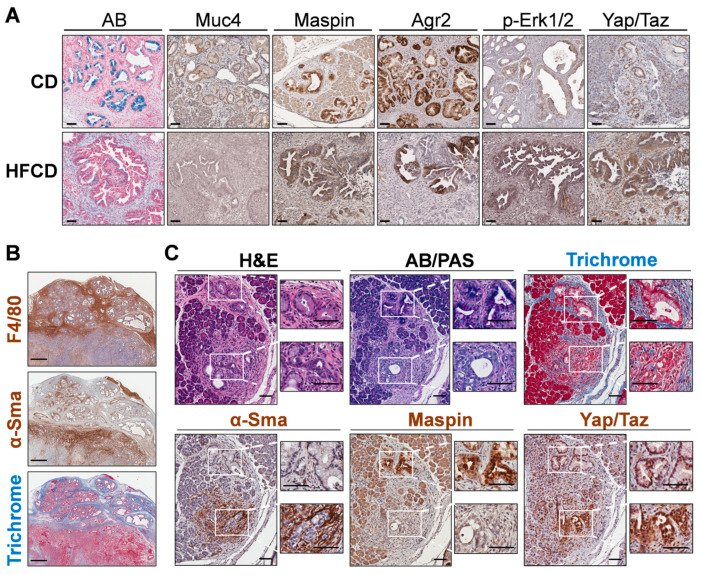
(**A**), differential immunohistochemistry (IHC) localization of distinct PDAC markers of classical PanIN in KC mice. Representative staining for Alcian Blue (AB), Maspin, Muc4, and Agr2 marked the cells lining PanIN, and nuclear p-Erk and Yap/Taz marked both PanIN and stromal cells in CD-fed KC mice (upper panels). In more advanced tumor areas (lower panels; HFCD-fed obese mice), some glands exhibited dramatically diminished AB, Maspin, and Muc4 and patchy Agr2 staining. Also, p-Erk and Yap/Taz staining remained in a PanIN/glandular epithelial and stromal pattern and was intensified by DIO. (**B**), areas with thick bands of Masson’s Trichrome-stained (TRI) collagen corresponded closely to F4/80+ macrophage staining, and only to a lesser extent with α-Sma+ CAFs. (**C**), appearance of PanIN and nearby atypical flat lesions (AFLs) viewed adjacent to PanIN in a CD-fed KC mouse at 9 mo. Epithelial cells lining PanIN were AB+, Maspin+ and associated with a limited amount of α-Sma+ CAFs. In contrast, the disorganized epithelial cells of AFLs were AB/PAS-, Maspin-, and surrounded by α-Sma+ CAFs. Scale bars: (**A**), 50 μm; (**B**), 500 μm; (**C**), 100 μm.

**Figure 6 cancers-16-01593-f006:**
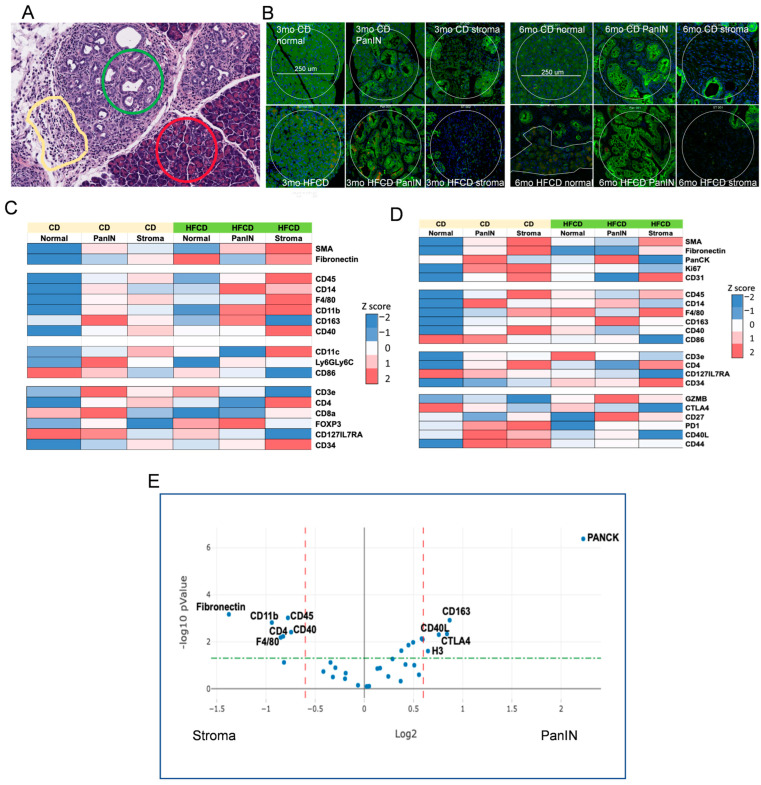
NanoString GeoMx^®^ Digital Spatial Profiling of DIO-induced immune cell subtype enrichment in stromal and PanIN areas in KC mice. We analyzed pancreas tissues from CD- and HFCD-fed KC mice at 3 mo and 6 mo (*n* = 3 mice/group). We collected 8 ROIs per slide, total of 20 normal, 37 stromal, and 39 PanIN data points, and applied GeoMx^®^ Housekeeping and controls, immune cell profiling, immune cell typing, and immune activation status antibody panels. (**A**), representative H&E staining at 6 mo at 100 μm scale, to illustrate selected normal (red circle), PanIN (green circle), and stromal (yellow irregular) ROIs. (**B**), IF staining for epithelial cell pan-cytokeratin- (green), immune cell CD45- (yellow), and cell nuclei (blue)-guided ROI selection of normal, PanIN, or stromal areas as in (**A**). Scale bar, 250 μm. (**C**,**D**), heat maps of significant DIO-induced changes at (**C**), 3 mo; (**D**), 6 mo. Diet drove many important shifts in immune distribution at 3 mo. (**E**), volcano plot of PanIN (Pan) vs. stromal (St) regions. Stromal areas attracted immune cells (CD45) including myeloid cells (F4/80) and myofibroblasts (α-Sma). In contrast, PanIN area preferentially attracted immunosuppressive M2 macrophages (CD163). Dashed lines indicate cutoffs for significance (green horizontal, *p* = 0.05) and fold change (red vertical, plus or minus 1.5-fold) levels.

## Data Availability

The proteomic and phosphoproteomic data were deposited to the MassIVE repository with the dataset identifiers MSV000094126 and MSV000094123, respectively.

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
