# Peer review of "Upregulated Matrisomal Proteins and Extracellular Matrix Mechanosignaling Underlie Obesity-Associated Promotion of Pancreatic Ductal Adenocarcinoma"

_cancers, 2024, doi:10.3390/cancers16081593_

Round 1
Reviewer 1 Report
Comments and Suggestions for Authors
The manuscript investigates the mechanisms of Diet-Induced Obesity (DIO)-induced pancreatic ductal adenocarcinoma (PDAC) in KC mice through comprehensive proteomic and phosphoproteomic analyses. They discovered significant enrichment of matrisomal proteins under DIO conditions, impacting ECM-receptor signaling pathways linked to tumorigenesis. They also noticed sex differences in matrisomal protein expression, suggesting delayed DIO-induced tumorigenesis in female KC mice. Additionally, they examined metabolic changes, glycolytic gene expression, and lipid synthetic gene expression under DIO conditions. Phosphoproteomic analysis reveals sites associated with mechanoregulatory signaling, shedding light on tumorigenic signal transduction pathways. They found the correlations between precursor lesions (PanIN and AFL), myCAF, macrophages, and collagen deposition in DIO-treated KC mice highlight key events in PDAC development. Overall, the study emphasizes the importance of matrisomal proteins and provides new insights into the mechanisms of DIO-induced PDAC tumorigenesis.
It would be beneficial to conduct functional assays to validate the functions of identified matrisomal proteins in tumorigenesis.
Considering sex differences, additional clarification is needed for Fig 2B and 2E. It is essential to specify whether these figures present combined results from both male and female subjects or if the data is separated by sex. Moreover, it would be advantageous to explore and present findings related to sex differences in both proteomic and phosphoproteomic analyses.
Author Response
Response to reviewer #1:
We thank the reviewer for their astute summary and comments. We have revised the manuscript taking their comments into account and have made the following specific revision and additions:
The comment about functional assays is well taken and we agree completely as to the value of functional assays to reinforce our conclusions. With this in mind, we offered a statement echoing this viewpoint in the text; this can be found in the modestly expanded second paragraph of the discussion. Also in this space, we decided not to include, but to make reference to data we have obtained that address the functional aspect of the processes involved through use of a novel protein inhibitor of ECM-receptor signaling. We hope that this brief mention of the direction of these data will suffice, as actually including them should require description of a new cohort of animals, slightly distinct feeding, timelines and procedures and would tend to bog down the methods section and detract from its clarity. As such, the results obtained do tend to reinforce the functional connections we provisionally concluded to be present in the present report. Thus, we hope that their current form is acceptable.
Another question was raised in the form of a request to clarify some of the information presented in Fig. 2. We recognize the deficiency in our explanations and corrected our views to make them more explicit. This was addressed by moving a small part of text from the results section and beginning the Discussion section with a new, concise paragraph on this point. In this paragraph, we explain that the data were indeed combined by sex because the markers of mucinous PanIN and accumulation of intra- and interlobular collagen did not differ much by sex; to prove this would have required more tissues per sex, whereas combining them provided the means to show the large effect with significance. Overall, the data agree with the previous published work (from which the whole cohort of tissues derives), showing a similar set of endpoints with slightly different approaches. Thus, these data both reinforce and clarify the previous data for the field, bringing in an understanding of the findings for some who might use these same techniques in their own work and thus, inherently recognize their value as well as their limitations. In particular, the albeit minor novel aspect we underscore is that the markers do not simply increase monotonically but occur transiently for reasons we probably do not yet fully understand. We hope these changes achieve the intended result and thank the reviewer for suggesting this be clarified.
This reviewer’s third and final comment seemed to us to be a request to delve into and “present” more findings already documented but not yet mentioned in the manuscript. We were very happy to address this and appreciate the opportunity to flesh out the data more completely to reinforce and improve the paper. Here, we expanded the third paragraph of the discussion, bringing to light new published reports emphasizing roles in PaCa and differential expression by sex for two proteins we found elevated more in males than females at 6 mo; Lcn2 and Lmo7. To provide a phosphoprotein result that differed between sexes, we found that the progesterone receptor was phosphorylated at Ser181 from early times and declined later, and this was driven by data in the females. Investigating the significance, we found that this event is linked to ERa activation, which could have some sex specific consequences in PDAC (although we are not presently cognizant of data to this effect). Thus, we included a brief description of these proteins as we felt was requested. With these changes, we hope that this reviewer will find the manuscript acceptable for publication.
Reviewer 2 Report
Comments and Suggestions for Authors
Simply said, it was unclear what was newly discovered in this experiment. The explains for each result with discussion are too long for readers if the results are not new finding. Therefore, it is hard to understand what is new in this result. Please cut the sentence in the Result section and the detailed explanations are moved to Discussion.
1. I wonder how significant 3 months faster of PDAC progression for the mice. Do you check whether the weight changes, weakness, etc. due to PDAC onset.
2. Is this mechanism exactly the same as the mechanism by which normally fed mice develop PDAC 3 months later? For example, the difference between HFCD and CD in Figure.3 is evaluated at the same 6 months. Did you check whether the protein expression pattern is similar at 6 months for HFCD and 9 months for CD.
3. In Figure 2 B, E., the images are displayed separately for Female and Male, but the mean value is finally displayed in B. On the other hand, the Masson Trichome is sex-neutral. On the other hand, the Masson Trichome is not sex-disaggregated. Why?
4. As far as I can see, the expression distribution of HFCD-Stroma (3mo) in Figure.6 C and CD-Stroma (6mon) in D are very similar, and Cd163, which you are focusing on, is higher in CD-stroma of D. If high Cd163 is close to PanIN, then 3 If Cd163 is higher in 3month HFCD-stroma, it is understandable that Cd163 is lower in 3month HFCD-stroma, but Cd163 is higher in 6month CD-normal than in 3month CD-normal, so does this mean PanIN?
Author Response
Response to reviewer #2:
We thank the reviewer for their incisive comments and queries. We have revised the manuscript taking their comments into account and have made the following specific revision and additions:
The first comment expresses a general feel about the clarity of the presentation as compared to the novelty; we thank the reviewer for tasking us with improving this aspect and hope we have improved this balance. We felt this was a good suggestion and followed it to remove sentences from the results section that was a bit too lengthy, although we did not remove many as most sentences make reference to figures and tables and we felt it could imbalance the paper to move too much into the discussion part, which we did expand by one paragraph and two other paragraphs were made longer.
In the reviewer’s first point, they bring into question an issue which is important in the field. It follows that, if obesity speeds up the rate at which humans get cancer, then obese people should acquire the disease at earlier ages. This reviewer appreciates the design and inherent limitations of animal experiments. The weight gain aspect of these experiments is challenging to document properly, as results from time to time can be unexpected and/or difficult to interpret. Mice gaining weight due to feeding a very rich cancer accelerating diet, at some point different for each mouse will plateau and either stop gaining or begin losing the weight, which may trigger compulsion to sacrifice the mouse as a “humane endpoint”. It is complicated, as social animals the weight loss can also have more “trivial” causes, like domination by another mouse in the same cage. Our experiments faced down these issues, made adjustments as needed and maintained the cohorts through the time points according to the prescribed plan. It would certainly be interesting to study the weight gain as a proxy to cachexia. However, new cohorts and a different response than the human endpoint would have to be in place from the beginning. We thank the reviewer for their valuable insights and understanding.
The reviewer’s second point and question prompted us to examine the proteomic data to evaluate whether the proteins elevated in CD fed KC mice were similar to those elevated at 6 mo by DIO. When we did this, we found there were 11 proteins elevated in both the CD fed mice and DIO mice at 6 mo and 19 elevated exclusively in the CD fed mice. In other words, there was a partial overlap. So, some proteins were the same and some also different. We noticed that Col6a1 and Col6a2 were two of the shared elevated proteins. In the recent literature, we found a paper that hypothesizes these two proteins are elevated by reverse mechanosignaling, unlike other matrisomal proteins, which are induced by stiff matrices. We wrote about these findings in the paper, adding this information to the discussion and thank the reviewer for prompting us to improve the paper in this manner.
Another question was raised in the form of a request to clarify some of the information presented in Fig. 2. We recognize the deficiency in our explanations and corrected our views to make them more explicit. This was addressed by moving a small part of text from the results section and beginning the Discussion section with a new, concise paragraph on this point. In this paragraph, we explain that the data were indeed combined by sex because the markers of mucinous PanIN and accumulation of intra- and interlobular collagen did not differ much by sex; to prove this would have required more tissues per sex, whereas combining them provided the means to show the large effect with significance. Overall, the data agree with the previous published work (from which the whole cohort of tissues derives), showing a similar set of endpoints with slightly different approaches. Thus, these data both reinforce and clarify the previous data for the field, bringing in an understanding of the findings for some who might use these same techniques in their own work and thus, inherently recognize their value as well as their limitations. In particular, the albeit minor novel aspect we underscore is that the markers do not simply increase monotonically but occur transiently for reasons we probably do not yet fully understand. We hope these changes achieve the intended result and thank the reviewer for suggesting this be clarified.
This reviewer’s fourth and final comment had to do with the findings on GeoMX. It was true that the 3 mo HFCD and 6 mo CD were rather similar. What we found more striking about the data was that just about every case the Cd163 was elevated in the PanIN. Indeed, comparison of stroma and PanIN directly in Fig. 6E showed this relation held true and significant across all the data, thereby associating this alternatively activated macrophage marker with the PanIN areas. We hope this is clear in the data, and we thank the reviewer for their interest and focus on this aspect of the paper. With the addition of the changes mentioned above, we hope that this reviewer will now find the manuscript acceptable for publication.
Reviewer 3 Report
Comments and Suggestions for Authors
General Statement: This is a manuscript detailing a temporal demonstration of protein/phosphoproteins as well as immune cells involved in high-fat/high-calorie fed KC mice in comparison to control diet. Histological evidence and phospho-, proteomic analyses with systems biology approach were utilized to depict potential molecular events that are distinctive in the obesity-related disease process in PDAC. Matrisomal proteins were demonstrated to be differentially expressed in HFCD vs CD fed KC mice. Immune cell involved in phases of PDAC development were characterized using spatial analyzer.
Major Issues: None seen with the broad approach utilized in this study; demonstrated potential targets or markers for further research in pancreatic carcinogenesis.
Minor Issues:
· Figure 6, legend (for ‘C, D, E’) – correct labeling.
· Check on clarity and style recommended (minimal).
Author Response
We thank the reviewer for their brief summary and kind words. As there were no major questions to address, we simply adjusted the labeling in the legend to Fig. 6 and had no other tasks to report. A few changes will be recognized, some of which may address issues of clarity and style as requested. We hope that the paper with its additional revisions will remain acceptable for publication for this reviewer and thank them once again for their assistance with this submission.